# Seeing Eye to AI: Human Alignment via Gaze-Based Response Rewards for Large Language Models

Angela Lopez-Cardona[1,2], Carlos Segura[1], Alexandros Karatzoglou[3], Sergi Abadal[2], and Ioannis Arapakis[1]

[1]Telefónica Scientific Research, Barcelona, Spain
[2]Universitat Politècnica de Catalunya, Barcelona, Spain
[3]Amazon, Barcelona, Spain
[1]{angela.lopezcardona, carlos.seguraperales,
ioannis.arapakis}@telefonica.com,
alexandros.karatzoglou@gmail.com, abadal@ac.upc.edu

## Abstract

Advancements in Natural Language Processing (NLP), have led to the emergence of Large Language Models (LLMs) such as GPT, Llama, Claude, and Gemini, which excel across a range of tasks but require extensive fine-tuning to align their outputs with human expectations. A widely used method for achieving this alignment is Reinforcement Learning from Human Feedback (RLHF), which, despite its success, faces challenges in accurately modelling human preferences. In this paper, we introduce GazeReward, a novel framework that integrates implicit feedback – and specifically eye-tracking (ET) data – into the Reward Model (RM). In addition, we explore how ET-based features can provide insights into user preferences. Through ablation studies we test our framework with different integration methods, LLMs, and ET generator models, demonstrating that our approach significantly improves the accuracy of the RM on established human preference datasets. This work advances the efforts to optimise AI alignment with human values, and explores the potential of cognitive data to shape future NLP research.

## 1 Introduction

Recent advancements in Natural Language Processing (NLP) have led to the emergence of Large Language Models (LLMs) like GPT (OpenAI, 2023), Llama (Dubey et al., 2024), Claude (Anthropic, 2024), and Gemini (Team et al., 2024), which excel across a range of tasks. These models, often consisting of billions of parameters, are trained on massive datasets and typically require extensive fine-tuning to align their outputs with human expectations [1]. Several works have focused on refining the way LLMs interpret and respond to user intent (Wang et al., 2023b), which has led to the development of novel alignment techniques. A common approach to achieving human alignment involves leveraging explicit human feedback as preference information. Currently, the most widely adopted method is Reinforcement Learning from Human Feedback (RLHF) (Ouyang et al., 2024). RLHF has been implemented in many state-of-the-art LLMs (Cui et al., 2024; OpenAI, 2023; Bai et al., 2022b), and has been shown to help align models to human instructions and mitigate the generation of toxic or harmful content (Kiegeland et al., 2024). However, a persistent challenge with this approach is the difficulty of acquiring sufficient high-quality training data (Casper et al., 2023).

To be able to capture the complexities of real-world user instructions, there is a need for meticulously handcrafted data (Wang et al., 2023b), which are resource-expensive and difficult to scale (Yang et al., 2023). Obtaining high-quality feedback from human annotators, usually provided after

---

[1]LLMs that are trained only on extensive datasets for language modeling are referred to as "pre-trained" LLMs. Subsequent approaches, such as human alignment, are categorized as "post-training".

examining a model response, suffers from several caveats (Casper et al., 2023). For instance, low inter-annotator agreement can result in inconsistent evaluations of the same model output due to varying interpretations, domain expertise, or biases. Moreover, "scalable oversight" – the ability to supervise models effectively with limited resources (Amodei et al., 2016) – remains an open problem. Inconsistent data quality is another issue, as cost-quality tradeoffs often arise when collecting human feedback.

To address these challenges, researchers have increasingly turned to LLMs as a form of AI-driven feedback, referred to as Reinforcement Learning from AI Feedback (RLAIF) Bai et al. (2022b). This method offers improved scalability, easier data collection, and cost-efficiency compared to traditional human-driven approaches (Bai et al., 2022b; Wang et al., 2023a; Madaan et al., 2023). However, it remains unclear what type of feedback signals, or a combination of feedback mechanisms, is optimal to align LLM with human goals (Casper et al., 2023). More research is needed to explore the underlying beliefs and expectations of human users (Casper et al., 2023), and how these can be incorporated into human alignment techniques. Furthermore, the alignment success of a language model is dependent on the quality of the underlying RM (Pace et al., 2024). Various alignment methods, such as RLHF, RLAIF, and Direct Preference Optimization (DPO) (Rafailov et al., 2023), rely on RM to incorporate feedback. Reward modelling is also essential for generating synthetic data for preference alignment and is often used in LLM inference to evaluate model outputs in techniques such as best-of-N sampling (Cui et al., 2024).

In this work, we propose a novel approach that incorporates Eye-tracking (ET) as an additional signal to address the challenge of human alignment. ET measures oculomotor behavior i.e. the movements and fixations of the eyes, which offers insight into visual attention and information processing (Kleinke, 1986; Land & Furneaux, 1997). This allows researchers to correlate observable eye movement patterns with underlying cognitive and perceptual processes during reading and language comprehension tasks (Kleinke, 1986; Krasich et al., 2018). Moreover, ET – unlike other (explicit) forms of feedback (e.g., questionnaire data, data annotation) – does not suffer from human biases, and offers a better temporal and spatial resolution (Zhang & Hollenstein, 2024). Several studies have shown a strong correlation between human eye movements and attention patterns in transformer-based models (Wang et al., 2024a; Bensemann et al., 2022; Sood et al., 2020a). Incorporating ET data into NLP tasks has also proven valuable, as demonstrated by numerous works (Huang et al., 2023; Khurana et al., 2023; Hollenstein et al., 2019; Yang & Hollenstein, 2023; Kiegeland et al., 2024; Deng et al., 2023a; Mathias et al., 2018; McGuire & Tomuro, 2021). Recently, Kiegeland et al. (2024) proposed the integration of ET in controlled sentiment generation to create a dataset that can be used in human alignment methods.

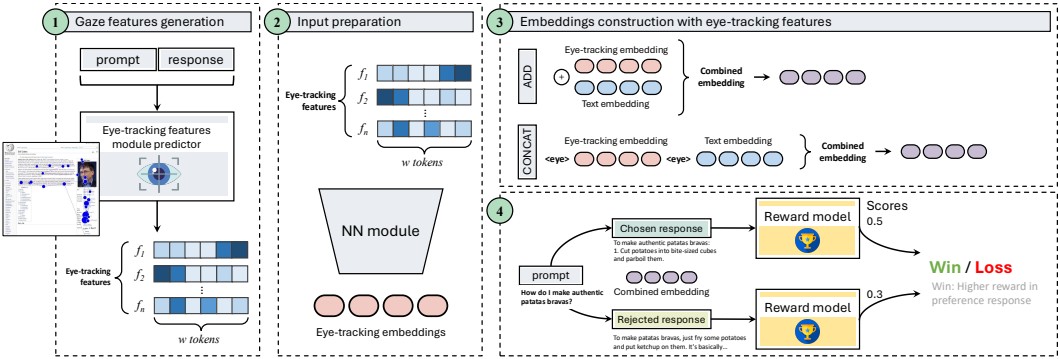

Figure 1: GazeReward Framework for using eye-tracking data for reward modelling. We use a generator model to compute ET features on a preference dataset $D$ and we train the human preference by combining both text and ET embeddings (See section 4 for details.)

Human alignment remains one of the biggest challenges in the development of LLM, with RM playing an important role in addressing this issue. This paper investigates how behavioural signals, particularly ET, can be operationalised as implicit feedback to improve human alignment (proposed approach is shown in Figure 1). Furthermore, we explore the use of ET prediction models that can

generate – automatically and with little effort – ET features in response to text input, which makes our solution not only cost-effective but also highly scalable. Our contributions are the following:

- We propose **GazeReward**, a novel and scalable framework that integrates implicit feedback in the form of ET data into the RM, a key component in modeling human preferences.

- We perform for the first time an ablation study that examines several state-of-the-art LLMs, various ET prediction models, and methods for incorporating ET features into the RM.

- We demonstrate experimentally substantial performance improvements with the GazeReward framework, showing accuracy gains of over 10% in RM predictions across diverse human preference datasets.

## 2 PRELIMINARIES

### 2.1 LARGE LANGUAGE MODELS-HUMAN ALIGNMENT

LLMs-Human Alignment typically involves training LLMs [2] on datasets curated by humans (learning from human feedback data) (Ouyang et al., 2024). This can be achieved through Supervised Fine-Tuning (SFT), where the model is trained on pairs of prompts ($x$) and corresponding human-generated responses ($y$) (Liu et al., 2024). Alternatively, alignment can be pursued via preference optimization, using a human preference dataset that differentiates between a better response ($y_w$) and a worse one ($y_l$) for the same prompt ($x$): $\mathcal{D} = \{(x^{(i)}, y_w^{(i)}, y_l^{(i)})\}_{i=1}^N$.

To this day, RLHF (Ouyang et al., 2024) remains the most popular technique used in state-of-the-art LLMs like GPT-4 (OpenAI, 2023), Claude (Bai et al., 2022b), Bard (Google, 2023), and Llama 2-Chat (Touvron et al., 2023). Different implementations of RLHF can vary in terms of data collection, training processes, and choice of RL algorithms. Typically, RLHF (Ouyang et al., 2024) involves three main steps: (1) collecting feedback, (2) training a RM based on that feedback, and (3) optimising the LLMs using RL techniques, such as Proximal Policy Optimization (PPO) Schulman et al. (2017). Since RLHF was first introduced, several advancements have been made, including fine-grained reward systems (Bai et al., 2022b; Wu et al., 2023b; Dong et al., 2023b; Wang et al., 2023c; 2024c), or replaced the original PPO algorithm with other RL techniques (Wu et al., 2023a).

An alternative to RLHF is DPO (Rafailov et al., 2023), which employs an offline RL approach to optimize language models based on preference data, without the need for a separate RM. While DPO can be used independently, it is often complementary to other training methods like SFT or statistical rejection sampling, to further improve human alignment based on a RM (Zhao et al., 2023; Liu et al., 2024; Dubey et al., 2024). Statistical rejection sampling, also called best-of-N or top-k-over-N (Bai et al., 2022b; Touvron et al., 2023; Dubey et al., 2024) is another widely used technique. Moreover, certain methods perform human alignment without RL to avoid instabilities, and fine-tune the model on filtered samples by a RM, or other sources (Dong et al., 2023a; Yuan et al., 2023).

A major challenge in human alignment techniques is data acquisition (Casper et al., 2023). This includes issues such as evaluator misalignment, supervision difficulties, and feedback quality (Casper et al., 2023). However, as AI systems continue to improve, LLMs are increasingly employed for tasks traditionally handled by humans, such as data annotation and generation. Unlike human feedback, AI-generated feedback offers better scalability, enabling faster and more cost-effective data collection. For example, RLAIF, introduced by Bai et al. (2022b), is a promising approach that trains reward models based on preferences generated by off-the-shelf LLMs. Variations of RLAIF have been explored in several studies (Lee et al., 2023; Jiao, 2023; Cui et al., 2024; Li et al., 2024; Yang et al., 2024). In the context of self-generating instructions, approaches like Self-Instruct (Wang et al., 2023a), Self-Refine (Madaan et al., 2023), and Self-Alignment (Li et al., 2023) demonstrate how models can autonomously generate datasets based on their learned human preferences.

Different alignment methods like RLHF and RLAIF rely on the RM to incorporate the human feedback. The RM learns to predict human preference based on labeled examples, serving as a proxy for human judgment later. Therefore, the success of language model alignment relies heavily on the quality of the underlying reward model (Pace et al., 2024), which in turn dictates the behaviour of

---

[2]Before the process of human alignment, these models are referred to as "pre-trained" LLMs.

Table 1: Eye-tracking (ET) features computed per word.

| Feature | Definition |
|---|---|
| First Fixation Duration (FFD) | Time spent on the initial fixation |
| Go-Past Time (GPT) | Cumulative fixation time before moving to the right |
| Total Reading Time (TRT) | Overall time spent fixating on a word |
| Number of Fixations (nFix) | Number of fixations on each word |
| Proportion of participants (fixProp) | Proportion of participants that fixated on the word |

the resultant chatbot (Shen et al., 2023). Even in LLM inference, methods like best-of-N sampling use the RM to evaluate model outputs (Cui et al., 2024). RM has also become crucial for generating synthetic data for preference alignment. In recent RLAIF methods, reward modeling has expanded beyond its traditional role and is now used to generate artificial feedback.

### 2.1.1 REWARD MODELING

In the original implementation (Ouyang et al., 2024), the goal of RM training is to train a classifier that predicts the probability of human preference $p^*$ between two completions (Equation 1), modelled by a Bradley-Terry model (Bradley & Terry, 1952). The typical setup involves showing two completions, with preferences being measured using win-loss-tie outcomes or a Likert scale to capture the strength of preference (Bai et al., 2022a). The data is processed into prompt-chosen-rejected trios, where the chosen completion, $y_w$, is preferred over the rejected one, $y_l$, forming the basis for training (Ouyang et al., 2024).

$$p^*(y_w \succ y_l \mid x) = \frac{\exp(r^*(x, y_w))}{\exp(r^*(x, y_w)) + \exp(r^*(x, y_l))}. \tag{1}$$

### 2.2 EYE-TRACKING

Eye-tracking (ET) systems monitor oculomotor behavior, such as eye movements and fixations, offering valuable insights into visual attention, information processing, and expands our understanding of reading and language comprehension. (Zhang & Hollenstein, 2024). Specifically, ET data often include fixations – pauses in eye movement to focus on specific areas (Mathias et al., 2020); saccades – rapid movements between two points (McGuire & Tomuro, 2021); scanpaths – sequences of fixations that reveal saccades and regressions (Yang & Hollenstein, 2023); and other temporal and spatial gaze behavior features (Zhang & Hollenstein, 2024). Incorporating ET data into NLP tasks often involves the use of several features listed in Table 1.

While several publicly available datasets such as ZUCO (Hollenstein et al., 2020b), ZUCO2 (Hollenstein et al., 2018), PROVO (Luke & Christianson, 2018), ETSA-I (Mishra et al., 2016a), ETSA-II (Mishra et al., 2018), GECO (Cop et al., 2017), GECO-MT (Colman et al., 2022) are widely used in ET research, obtaining real ET data for NLP tasks remains a challenge. This is primarily due to the cost and precision requirements of ET equipment, the unavailability of gaze data during inference, as well as privacy concerns (Khurana et al., 2023). To address these challenges, two main approaches have been proposed. The first involves integrating ET data into the model during training through methods like Multi-task learning (MTL), which eliminates the need for ET data during inference (Mishra et al., 2018; Klerke et al., 2016; Ren & Xiong, 2021; Yu et al., 2024; Deng et al., 2024). The second approach involves techniques that directly predict users' gaze behaviour (Deng et al., 2024; 2023a; Zhang & Hollenstein, 2024; Wang et al., 2024a), creating synthetic ET data for any given text or stimulus (Deng et al., 2023b; Bolliger et al., 2023; Khurana et al., 2023; Li & Rudzicz, 2021; Hollenstein et al., 2021; 2022).

## 3 RELATED WORK

**Reward Modelling.** The most popular approach to reward modeling follows the framework introduced by Ouyang et al. (2024). Several studies have examined alternative versions for refining RMs. For instance, Bai et al. (2022b) proposed more fine-grained reward structures, evaluating helpfulness

and harmlessness separately. Other approaches have explored different reward modelling strategies (Wu et al., 2023b; Dong et al., 2023b; Wang et al., 2023c). Another line of research has focused on Process Based Reward Models (PRMs) (Lightman et al., 2024; Uesato et al., 2022) which differ from conventional RMs by predicting the correctness of intermediate steps, rather than solely evaluating final outputs. Other studies implement data augmentation techniques (Shen et al., 2023), or cross-attention mechanisms between encoded input text and candidate pairs (Jiang et al., 2023b). Moreover, some works have leveraged synthetic preference data for reward modelling (Cui et al., 2024; Jiao, 2023). Wu et al. (2024b) built upon the LLM-as-a-Judge framework Zheng et al. (2023) by introducing LLM-as-a-Meta-Judge, which evaluates the model's judgments to generate preference pairs that enhance its decision-making capabilities. Finally, Pace et al. (2024) incorporated a self-training approach to improve reward model training. However, to date, no research has explored the integration of ET or other implicit feedback signals into RM.

**Eye-tracking in Natural Language Processing.** Several studies have investigated the use of ET data for a variety of NLP tasks, such as named entity recognition (Hollenstein & Zhang, 2019; Ren & Xiong, 2021; Yu et al., 2024; Hollenstein et al., 2019), text comprehension (Ahn et al., 2020; Reich et al., 2022; Sood et al., 2020b), language modelling (Huang et al., 2023; Huang & Hollenstein, 2023; Deng et al., 2023b), and question answering (Zhang & Hollenstein, 2024; Wang et al., 2024a). Other applications include code comprehension (Alakmeh et al., 2024), code summarization (Zhang et al., 2024) and hallucination detection (Maharaj et al., 2023). Eye-tracking has also been applied to sentiment analysis and sarcasm detection tasks (Mishra et al., 2016a;b; 2018; Barrett et al., 2018; Huang et al., 2023; Khurana et al., 2023; Hollenstein et al., 2019; Yang & Hollenstein, 2023; Kiegeland et al., 2024; Deng et al., 2023a; Mathias et al., 2018; McGuire & Tomuro, 2021). The most relevant work to our approach is by Kiegeland et al. (2024), which introduced a dataset generation method using ET signals for DPO, building on the controlled sentiment generation framework proposed by Deng et al. (2023a); Yang & Hollenstein (2023). While this study has contributed to the first steps towards integrating ET for human alignment in LLMs, it is task- and dataset-specific, often relying on ranking criteria that underutilize the potential of ET feedback. In contrast, our approach presents a more general framework by directly incorporating implicit feedback into the RM, rather than limiting its application to dataset creation.

## 4   GAZEREWARD: REWARD MODELING WITH ET FEEDBACK

In this section, we discuss the proposed framework for augmenting the RM using implicit feedback derived from ET signals (Figure 2). Initially, we generate the ET features (subsection 4.1) considering two state-of-the-art ET prediction models. Next, we combine the ET features with the text (subsection 4.2), producing different types of combined embeddings, and finally pass them as input into the RM to obtain the reward for the prompt and its corresponding response (subsection 4.3).

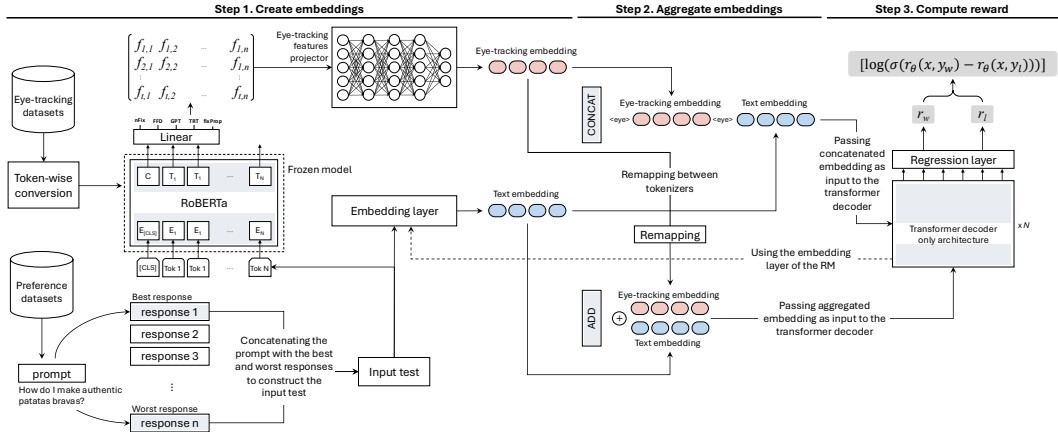

Figure 2: Overview of the **GazeReward** framework, incorporating eye-tracking features into the reward model. The architecture is illustrated in the figure using the second ET prediction model, but it would be identical if the first one were used instead (see subsection 5.1)

## 4.1 EYE-TRACKING FEATURES GENERATION

As discussed in subsection 2.2, obtaining organic ET features for NLP applications presents several challenges. In this work, we consider an approach inspired by RLAIF research, where feedback is artificially generated from pre-trained LLMs and, in particular, from ET prediction models. Specifically, we incorporate the output of two different ET prediction models (Li & Rudzicz, 2021; Huang & Hollenstein, 2023) and evaluate the impact of different set of features. As input to these models, we pass the same text as we do in the RM: a combination of prompt $x$ and response $y$. The output is a set of ET features, denoted as $f_{et}$, for each token $f_{et} = \{f_1, f_2, \ldots, f_w\} \in \mathbb{R}^{w \times f}$, where $w$ represents the number of tokens in the tokenizer used by the ET prediction model, and $f$ is the number of features. Depending on the specific model, between one and five synthetic features $f_{et} = \{f_1, f_2, \ldots, f_w\} \in \mathbb{R}^{w \times f}$ are generated per token for the input text.

## 4.2 RM AUGMENTATION USING EYE-TRACKING FEATURES

We implement two different approaches for incorporating ET features into the RM, as shown in Figure 2. In the first approach, **GazeConcat**, we concatenate the ET embeddings with the text embeddings. In the second approach, **GazeAdd**, we add the ET embeddings to the text embeddings. Furthermore, we concatenate the prompt and the response to be evaluated and pass them through the pre-trained embedding layer of , to generate the embeddings $H = \{h_1, h_2, \ldots, h_n\} \in \mathbb{R}^{n \times d}$, where $n$ is the number of tokens in the tokenizer used by the RM and $d$ is the model embedding size.

To project these features to the model embedding size ($d$), we use a Multilayer Perceptron (MLP) ET feature projector $fp()$. The $fp()$ consists of two linear layers, two dropout layers, two Layer Normalization layers, and `ReLU` activation, designed for stable, non-linear ET feature representation and overfitting prevention. The model's input dimension dynamically adjusts to accommodate the number of features used during training. The ET features projector can be formulated as $emb_{ETF} = fp(f_{et}) \in \mathbb{R}^{w \times d}$ (Figure 2). This formula describes the projection of ETF features ($f_{et}$) through a function $fp$, resulting in an embedding matrix $emb_{ETF}$ with dimensions $w \times d$, where $w$ represents the number of tokens and $d$ the embedding dimension.

**GazeConcat:** The ET embedding, denoted as $emb_{ETF}$, is concatenated with the text embedding $H$ to form the input for the RM. To distinguish between the two modalities, we introduce two special tokens: ⟨eye⟩ and ⟨/eye⟩, which flag the start and end of the ET embedding, respectively (Figure 2). These special tokens are randomly initialized as one-dimensional vectors and added to the embedding layer or the RM model for training. The final input is structured as: $(emb(<eye>) \circ emb_{ETF} \circ emb(</eye>) \circ H)$. The same process is applied to the attentions masks.

**GazeAdd:** The input to the RM consists of the ET embedding $emb_{ETF}$ and the text embedding $H$, which are added in an elementwise fashion: $(emb_{ETF} + H)$. The two ET prediction models use different tokenizers, which also differ from those used by the base models in the RM. As a result, the number of tokens $n$ in the input for the RM and the number of tokens $w$ generated by the ET prediction model may not match. To address this embedding alignment issue, and have the same dimension, we remap the ET features from the $w$-token space to the $n$-token space used by each base model in the RM. Further implementation details can be found in Appendix A.1.3.

Table 2: Overview of different corpora used in the study to train the reward model.

| Corpus | Train set | Val. set | Test set | Lang. | Reference |
|---|---|---|---|---|---|
| OASST1 | 6567 | 1160 | 416 | EN* | Köpf et al. (2023) |
| HelpSteer2 | 5938 | 1049 | 364 | EN | Wang et al. (2024c) |

## 4.3 REWARD MODEL

The RM's architecture and hyperparameters are identical to those of the pretrained LLM, except that the classification head used for next-token prediction is replaced with a regression head that outputs a scalar reward (Touvron et al., 2023). This scalar reward indicates the quality of the model generation, corresponding to the predicted score for the final reply in a conversation. Differences in

these rewards represent the log-odds that one response is preferred over another. The loss function is defined in Equation 2, where $y_w$ refers to the preferred response in a pair of completions $y_w$ and $y_l$. The dataset $D$ consists of human comparisons, where $r_\theta\left(x, y_w\right), r_\theta\left(x, y_l\right)$ represents the RM $\theta$ scalar outputs for the preferred and less preferred completions, respectively Ouyang et al. (2024).

$$\text{loss}(\theta) = -E_{(x, y_w, y_l) \sim D}\left[\log\left(\sigma\left(r_\theta\left(x, y_w\right) - r_\theta\left(x, y_l\right)\right)\right)\right] \tag{2}$$

In the proposed method, we augment the traditional RM, which uses text input (a combination of the prompt $x$ and response $y$), by incorporating (artificial) implicit feedback through ET features generated from the same text. These ET features provide valuable information for capturing human preferences, thereby improving the RM's performance.

## 5 EXPERIMENTS

### 5.1 EXPERIMENTAL SETUP

**Datasets.** For our experiments, we use the OpenAssistant Conversations dataset's (OASST1) (Köpf et al., 2023) and HelpSteer2 (Wang et al., 2024c) ( Table 2). OASST1 is a human-generated, human-annotated, assistant-style conversation, created through global crowdsourcing and widely used for human alignment tasks (Köpf et al., 2023; Dettmers et al., 2023; Wu et al., 2024b). We filtered all non-English text, as the ET prediction models were exclusively trained on English data. Among the different responses in the dataset, we selected the two most distinct responses to compare the chosen and the rejected responses (Wang et al., 2024b). HelpSteer2 is a more recent, English-only dataset that has been used in studies such as Wang et al. (2024b;c). The dataset provides annotations for five response attributes: helpfulness, correctness, coherence, complexity, and verbosity. To transform it into a preference dataset, we designate the response with the higher helpfulness score as the chosen response and the other as the rejected response, following a method similar to that used in DPO training (Wang et al., 2024c) (see Appendix A.1.1 for more details about the datasets).

**Dataset Preparation.** To tune LLMs for human-AI interaction, we need to define a chat dialogue protocol that allows the model to understand human instructions and rate them. To this end, we adopt a chat protocol that utilizes special header and termination tokens, similar to the format used in Llama 3. For example, in the case of the Llama 3 8B model, the concatenation of a prompt and its corresponding response would follow this structure: *<im_start>user* Example Prompt *<im_end>* *<im_start>assistant* Example Response *<im_end>* (see Appendix A.1.1 for more details).

**Models.** As RM base models we use the pretrained checkpoint of Hugging Face (Appendix A.1.4) for Llama 3 8B, Llama 3 8B-instruct (Dubey et al., 2024) and Mistral 7B (Jiang et al., 2023a).

**ET prediction models.** In our analyses, we utilise two state-of-the-art ET prediction models, both pre-trained to predict ET features and kept frozen in our implementation. The input to these models is the same text used for the RM, with minimal modifications (Appendix A.1.2). The first model (Huang & Hollenstein, 2023), consists of a T5 embedding layer (Raffel et al., 2020), a two-layer BiLSTM (Hochreiter & Schmidhuber, 1997), and a one-hidden-layer MLP. This model was trained on the Dundee, GECO (Cop et al., 2017), ZuCo1 (Hollenstein et al., 2018), and ZuCo2 (Hollenstein et al., 2020a) datasets, and predicts total reading time (TRT) per token (Figure 3). The second model (Li & Rudzicz, 2021), is based on RoBERTa (Liu et al., 2019) with a regression head on each token. This head is a linear layer that outputs five features: FFD, fixProp, GPT, TRT, and nFix (Table 1). The model is initialized with pre-trained weights and fine-tuned on the ZuCo1 (Hollenstein et al., 2018), ZuCo2 (Hollenstein et al., 2020a) and PROVO (Luke & Christianson, 2018) datasets. Since RoBERTa's maximum sequence length is 512 tokens and our input sequences are longer, we employ a sliding window approach. The input is split into 512-token segments with a 50-token overlap, and the results are combined using a linear weighted approach. Further details on these models and their integration into our framework are provided in Appendix A.1.2.

**Baseline models.** To evaluate the improvement in accuracy for a RM that incorporates implicit feedback, and specifically ET signals, we compare the same RM with and without the ET embeddings. For each dataset and model, we train and evaluate all combinations of integrating and combining ET features, and then compare them against a RM trained on the same base model and dataset but without implicit feedback.

**Evaluation metrics.** Performance is determined by measuring the model's ability to predict the better reponse from pairs of replies with different ranks. Accuracy is calculated as the percentage of cases where the reward score for the preferred response is higher than that of the less preferred response, based on a held-out dataset. This method follows similar approaches found in Touvron et al. (2023); Yuan et al. (2023); Köpf et al. (2023); Cui et al. (2024). We use the test split proposed by the authors for each dataset (Table 2). We also conduct a complementary evaluation on RewardBench (Lambert et al., 2024), a benchmark dataset created for evaluating performance and safety features of RM's (see Appendix B for more details).

**Training procedure.** In our implementetion the ET modules remains frozen (Figure 2). For the RM, we fine-tune the open-source models previously introduced with QLoRA (Dettmers et al., 2023) a Parameter-Efficient Fine-Tuning (PEFT) method based on Low-Rank Adaptation (LoRA) (Hu et al., 2021), with other memory optimization techniques. We follow the training process for the RM as outlined in Touvron et al. (2023); Ouyang et al. (2024). We independently train each model on its respective dataset for two epochs, as detailed in Wang et al. (2024c). For hyperparameter tuning, we reserve 15% of each dataset for validation (shown in Table 2). The best-performing checkpoints are selected based on the lowest validation loss and used for performance evaluation. We perform a grid search to determine the optimal batch size and testing values of $\{8, 16, 32\}$. The AdamW optimizer (Loshchilov & Hutter, 2019) is used, with the Learning Rate (LR) is tuned over the range $\{1, 5, 10, 50\}$e-6, following the values reported in Touvron et al. (2023); Cui et al. (2024); Wang et al. (2024c). Additionally, we evaluate different LR schedulers: constant, linear, and cosine with a minimum LR. Further hyperparameter values and implementation details for both the RM and the ET projector can be found in Appendix C.

## 5.2 RESULTS

The results of our experiments on the OASST and HelpSteer datasets, covering all possible combinations of ET features, models, and inclusion methods, as shown in Table 3 and Table 4 respectively. For the Mistral model, results for the **GazeAdd** method are unavailable due to the inability to map features between the ET prediction model's tokenizer and the reward model's tokenizer (details in Appendix A.1.3). In what follows, we present key findings based on three seeds, reporting the average, mean, and statistical significance.

Table 3: Reward modeling accuracy (%) for OASST1 dataset. The highest results are in bold and the second highest are underlined.

| | | Llama-3-8B-Instruct | | Llama-3-8B | | Mistral-7B | |
|---|---|---|---|---|---|---|---|
| | baseline | $65.9 \pm 0.5$ | diff (%) | $65.5 \pm 2.1$ | diff (%) | $66.3 \pm 0.1$ | diff (%) |
| **GazeConcat** | $fcomb_1$ | $69.0 \pm 0.4*$ | 4.7 | $69.3 \pm 0.6$ | 5.9 | $67.6 \pm 1.7$ | 2.1 |
| | $fcomb_{2.5}$ | $70.2 \pm 0.3**$ | 6.6 | $\mathbf{71.5 \pm 0.5}$ | 9.2 | $\underline{70.2 \pm 0.4*}$ | 5.9 |
| | $fcomb_{2.2}$ | $\underline{70.0 \pm 0.4**}$ | 6.3 | $\underline{71.2 \pm 0.8}$ | 8.8 | $\mathbf{71.0 \pm 1.0}$ | 7.1 |
| **GazeAdd** | $fcomb_1$ | $68.9 \pm 0.9$ | 4.6 | $68.9 \pm 1.0$ | 5.3 | - | |
| | $fcomb_{2.5}$ | $\mathbf{70.2 \pm 0.1*}$ | 6.6 | $69.5 \pm 0.3$ | 6.1 | - | |
| | $fcomb_{2.2}$ | $69.0 \pm 0.4*$ | 4.7 | $68.3 \pm 0.7$ | 4.4 | - | |

Table 4: Reward modeling accuracy (%) for Helpsteer2 dataset. The highest results are in bold and the second highest are underlined.

| | | Llama-3-8B-Instruct | | Llama-3-8B | | Mistral-7B | |
|---|---|---|---|---|---|---|---|
| | baseline | $54.7 \pm 0.7$ | diff (%) | $53.3 \pm 0.8$ | diff (%) | $54.1 \pm 0.3$ | diff (%) |
| **GazeConcat** | $fcomb_1$ | $\underline{61.1 \pm 1.2*}$ | 11.8 | $59.1 \pm 0.2*$ | 10.8 | $\underline{57.6 \pm 2.3}$ | 6.5 |
| | $fcomb_{2.5}$ | $58.5 \pm 1.6$ | 7.0 | $\underline{60.3 \pm 0.5**}$ | 13.2 | $\mathbf{58.7 \pm 2.4}$ | 8.5 |
| | $fcomb_{2.2}$ | $60.6 \pm 3.3$ | 10.9 | $57.9 \pm 2.0$ | 8.6 | $56.0 \pm 2.4$ | 3.4 |
| **GazeAdd** | $fcomb_1$ | $\mathbf{62.3 \pm 0.6**}$ | 13.9 | $\mathbf{62.4 \pm 1.0**}$ | 17.0 | - | |
| | $fcomb_{2.5}$ | $59.6 \pm 1.1*$ | 9.0 | $58.6 \pm 1.2*$ | 10.0 | - | |
| | $fcomb_{2.2}$ | $60.3 \pm 0.5**$ | 10.2 | $59.3 \pm 0.1*$ | 11.3 | - | |

**Effect of Model Initialization.** We evaluate the impact of model initialization on performance. Open-access LLMs typically come in two forms: a pre-trained version without human alignment and a final version that has undergone alignment with human feedback. Since we lack access to intermediate checkpoints, we experiment with both pre-trained models (Mistral 7B and Llama 3) and models that are already human-aligned (Llama 3 Instruct). Our goal is to confirm that our method is effective for RM initialized with both pre-trained and human-aligned checkpoints. When comparing accuracy improvements relative to the baseline (without ET features), all models show considerable gains from incorporating implicit feedback. Notably, the Llama 3 8B and Mistral 7B models, which had no prior alignment, demonstrate performance improvement from the incorporation of ET features, indicating that unaligned models can benefit from implicit feedback.

**Inclusion method.** The results shown Table 3 and Table 4 indicate that both **GazeConcat** methods and **GazeAdd** introduce a substantial performance improvements to the RM. Across both datasets, concatenating embeddings (**GazeConcat**) delivers more consistent results. Incorporating ET information through specialized separator embeddings allows the model to process both text and ET features more robustly. However, in the HelpSteer dataset (Table 4), directly adding ET information to the text embeddings (**GazeAdd**) results in the greatest improvement over the baseline.

**Eye-tracking (ET) feature importance.** Different ET features capture distinct aspects of reading behaviour and information processing, influencing model performance uniquely (Zhang & Hollenstein, 2024). Here, we examine how model performance varies when incorporating three different feature combinations generated by two different ET prediction models: $fcomb_1$ – TRT generated by the first ET prediction model; $fcomb_{2.5}$ – five features (nFix, FFD, GPT, TRT, fixProp) generated by the second ET prediction model; and $fcomb_{2.2}$ – TRT and FFD generated by the second ET prediction model. TRT and FFD are widely used in ET research (Huang et al., 2023; Huang & Hollenstein, 2023; Zhang & Hollenstein, 2024; Maharaj et al., 2023; Wang et al., 2022), and they have been shown to correlate with attention scores from pre-trained transformer models (Wang et al., 2024a; Bensemann et al., 2022; Sood et al., 2020a) and with gradient-based saliency(Hollenstein & Beinborn, 2021; Wu et al., 2024a). When comparing results, we observe that the RM benefits from implicit feedback regardless of the ET feature combination or ET prediction model used. Specifically, in most cases, $fcomb_1$ yields the best results, particularly with the **GazeAdd** method. For **GazeConcat**, $fcomb_{2.2}$ and $fcomb_{2.5}$ performs best in general. We attribute the superior performance of $fcomb_1$ to how the ET prediction model generating the fixations was trained, including the data and preprocessing methods used (see Appendix A.1.2). Moreover, when comparing $fcomb_{2.2}$ and $fcomb_{2.5}$ – both generated by the same model – only in one case does integrating nFix, GPT, and fixProp improves performance. In some instances, using $fcomb_{2.5}$ results in worse performance than the baseline, confirming findings provided by prior studies, which suggest that features related to reading time, such as FFD and TRT, contribute most to performance gains.

**RewardBench.** As a side contribution, we evaluate our best performing models (trained on the OAAST1 dataset) on RewardBench. This evaluation is not intended to directly compare our method with larger, more resource-intensive RM, but rather to show that through the integration of multimodal signals like ET features we can significantly enhance the performance of RM models. The results shown in Table 5 demonstrate consistent improvements as previously observed, with gains exceeding 40% for the Mistral model – a notable gain considering that the base RM is the same. We note that the performance of the baseline models is impacted by RM trained on base models with less than 9B parameters and on relatively small datasets (see details in Appendix B).

Table 5: Reward modeling accuracy (%) evaluating on RewardBench dataset. All models are trained on OASST1 dataset. The highest results are in bold and the second highest are underlined.

| | | Llama-3-8B-Instruct | | Llama-3-8B | | Mistral-7B | |
|---|---|---|---|---|---|---|---|
| | baseline | 46.9 | diff(%) | 50.9 | diff(%) | 41.2 | diff(%) |
| **GazeConcat** | $fcomb_1$ | 57.8 | 23.1% | 58.4 | 14.5% | 59.9 | 45.4% |
| | $fcomb_{2.5}$ | **58.4** | **24.4%** | 58.1 | 14.1% | 60.3 | 46.4% |
| | $fcomb_{2.2}$ | 58.1 | 23.8% | **58.5** | **14.8%** | **60.5** | **46.9%** |
| **GazeAdd** | $fcomb_1$ | 56.5 | 20.3% | 56.6 | 11.2% | - | |
| | $fcomb_{2.5}$ | 54.9 | 16.9% | 53.8 | 5.6% | - | |
| | $fcomb_{2.2}$ | 55.4 | 17.9% | 52.5 | 3.1% | - | |

## 6 DISCUSSION

In this work, we introduced a novel framework for integrating implicit feedback into the Reward Model, a key component for aligning LLMs and generating synthetic data for further alignment. We validated our approach using widely-adopted, open-source models such as Llama 3 and Mistral, for initializing the RM. By employing two different models to generate ET features, our results show that incorporating implicit feedback consistently improves the RM's ability to predict user preferences, regardless of the model used and without the need to reach large parameter counts or train on massive datasets. Additionally, our method leverages ET features generated by models, making it fully scalable and applicable to various human alignment methods, including those that involve artificially generated datasets. This work advances the ongoing discussion on optimizing AI alignment with human values and shows the potential of multimodal signals for NLP research enhancing current methods.

### 6.1 LIMITATIONS & FUTURE WORK

**Data**. A limitation of our study is that both ET prediction models were trained on a relatively small datasets (Appendix A.1.2) that are not tailored to our tasks. Future work could benefit from directly collecting ET data specifically for LLM-generated responses, to offer insights into human reading comprehension and information processing of prompts, which could further improve model performance. Additionally, since the ET prediction models used in our experiments were trained on English corpora, the method's generalizability to other languages requires further investigation. Moreover, we explored two methodologies for integrating ET features into the RM, but other approaches could prove more effective. For instance, ET features could be used to modify the RM's attention mask, as suggested by Zhang & Hollenstein (2024). Regarding dataset selection, both models used, Mistral 7B and Llama 3, were fine-tuned on publicly available data, though specific details on the datasets are limited. Therefore, we cannot discount the possibility that the datasets we used may have been part of the models' pretraining, particularly for Llama 3 7B Instruct, which has already undergone human alignment. However, as we compare against baselines using the same model checkpoints, any potential effects would be consistent across both conditions. Finally, it is important to note that although the ET prediction models remain fixed during training, our solution has a slightly higher number of parameters than the baseline.

**Training**. The scaling trends for the RM (Touvron et al., 2023) show that larger models or models trained on massive datasets perform better. A promising direction would be to validate our framework on larger models, without relying on PEFT methods, and on larger datasets. We are confident that, despite the considerable computational costs this may entail, our framework is capable of scaling effectively. Another direction is integrating the proposed RM into an alignment method like RLHF, or applying it in rejection sampling to generate synthetic preference datasets, ensuring that accuracy gains in the RM translate to improvements in the final LLMs.

### REPRODUCIBILITY STATEMENT

All the code necessary to reproduce is in the GitHub repository [3]. Both datasets used are publicly available (Köpf et al., 2023; Wang et al., 2024c). Additionally, both ET prediction models have been trained with public datasets (Cop et al., 2017; Hollenstein et al., 2018; 2020a).

### IMPACT STATEMENT

Since our research uses only synthetic ET data, there are no privacy concerns or need for large-scale experiments involving human subjects. We should also raise attention to the limitations of human feedback and ET prediction models bias, that inevitably reflect aspects of their training data.

### ACKNOWLEDGMENTS

This research is supported by Horizon Europe's European Innovation Council through the Pathfinder program (SYMBIOTIK project, grant 101071147) and by the Industrial Doctorate Plan of the De-

---

[3]https://github.com/Telefonica-Scientific-Research/gaze_reward

partment of Research and Universities of the Generalitat de Catalunya, under Grant AGAUR 2023 DI060. We also want to thank Sebastian Macaluso for his important feedback during the project.

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

# A APPENDIX

## A.1 IMPLEMENTATION DETAILS

This section provides further details on the implementation of our method. Subsection A.1.1 provides more details on the datasets used and their preprocessing steps. In subsection A.1.2, further information is given about the models used for generating ET features, along with the specific preprocessing required for each. Subsection A.1.3 explains the process of mapping the fixations from the tokenizer used by the generation model to the tokenizer used by the Reward Model. Subsection A.1.4 give more detials on the checkpoints used fot the RM backbone models. Finally, additional implementation details are discussed in subsection A.1.5.

### A.1.1 DATASET PROCESSING

In this subsection, we provide more details about the datasets used and the preprocessing to train the RM. We use two different datasets: OpenAssistant Conversations dataset's (OASST1) [4] (Köpf et al., 2023) and HelpSteer2 [5] (Wang et al., 2024c).

**OASST1**. A human-generated, human-annotated assistant-style conversation corpus consisting of 161,443 messages in 35 different languages, resulting in over 10,000 complete and fully annotated conversation trees. The basic data structure is a Conversation Tree (CT), with nodes representing written messages in a conversation. A CT's root node represents an initial prompt, given by the prompter. The data was collected using a web-app interface as a product of a worldwide crowd-sourcing effort involving over 13,500 volunteers, dividing the collection of each tree into five separate steps: prompting, labelling prompts, adding reply messages as prompter or assistant, labelling replies, and ranking assistant replies.

**HelpSteer2.** A CC-BY-4.0-licensed open-source helpfulness dataset, designed to train state-of-the-art RM consisting on 10,000 response pairs. It collects prompts mainly from ShareGPT [6], focusing on user inputs and filtering out non-English and programming-related prompts for quality. The prompts are clustered into topics and sampled based on complexity to ensure diversity. Multi-turn prompts are generated using an in-house model, with responses sourced from various internal models and human annotators. For each response, they annotate five attributes (helpfulness, correctness, coherence, complexity, and verbosity) on a Likert-5 scale involving multiple annotators for each response, ensuring high-quality ratings across five attributes.

**Conversation format and dataset preparation.**
To fine-tune LLMs for human-AI interaction, we need to define a chat protocol. We use a multi-message chat setup with a special header and termination tokens, similar to the one in Llama 3 Dubey et al. (2024). The header tokens differentiate the turns between the user and the system. For this, we use the *apply_chat_template*[7] feature from *FastTokenizers* in the *transformers* library.

The tokenizer used by the Meta-Llama-3-8B-Instruct model already incorporates this chat format since this model has already undergone human alignment. Therefore, we use this format in our experiments. For the other two models, we employ the default chat format provided by their respective tokenizers. We add new tokens in the embeddings layer for these chat formats and we train them as part of our process. Below, we provide an example of the template for each model.

- **Meta-Llama-3-8B-Instruct:** *<begin_of_text><start_header_id>user<end_header_id>* Example Prompt *<eot_id><start_header_id>assistant<end_header_id>* Example Response *<eot_id>*

- **Meta-Llama-3-8B:** *<im_start>user* Example Prompt *<im_end>* *<im_start>assistant* Example Response *<im_end>*

- **Mistral-7B:** *[INST]* Example Prompt *[/INST]* Example Response **

### A.1.2 EYE-TRACKER FEATURES GENERATION MODELS

Special tokens are removed from the text before it is tokenized with the corresponding tokenizer used for the ET generator model. This is done to ensure that special tokens related to the chat format are not included in the input and are not assigned ET features to them, since these tokens are just for the RM to understand the chat format.

**First model:** Model presented in Huang & Hollenstein (2023). The code for the model along with the weights is publicly available, so we used the pre-trained checkpoint and we adapted their code for our implementation. This model was trained on several eye-tracking datasets, including Dundee (Kennedy et al., 2012), GECO (Cop et al., 2017), ZuCo1 (Hollenstein et al., 2018), ZuCo2 (Hollenstein et al., 2020a). More detailed information about this datasets is presented in Table 7. The best model achieves an mean squared error (MSE) of 4.02 on a randomly held-out test set (25%

---

[4] https://huggingface.co/datasets/OpenAssistant/oasst1
[5] https://huggingface.co/datasets/nvidia/HelpSteer2
[6] https://huggingface.co/datasets/RyokoAI/ShareGPT52K
[7] https://huggingface.co/docs/transformers/main/en/chat_templating

of all data). This model has 17.5M pararemetes and remains frozen during training. In Figure 3 we show an example of the synthetic total reading time (TRT) generated for the chosen and rejected response to a prompt.

For training this model, since the fixation duration is distributed differently across corpora, the authors normalize the fixation duration for each corpus, by dividing it by the mean duration of the corpus. Moreover, they map the duration values to discrete space $[1, 2, , K]$. Using K-quantiles, the fixation values were partitioned into $K$ subsets of nearly equal sizes, and each value was assigned to the index of the corresponding subset. The model is then trained in a multi-task setting, computing the mean and variance of the fixation duration. This quartile-based processing is used in other works (Huang et al., 2023) that use ET data to improve performance in NLP tasks, and we believe it is part of the reason why we obtained better results with this combination when training the RM. The authors proposed a method specifically for converting word-level TRT to token-level fixation data during model training. Initially, the TRT of a word is assigned to its characters, then a small number is assigned to the last character of the word (mainly to give small values to punctuation). After tokenizing the word the span of each subword is obtained, and the maximum value in each span is taken as the final token-level fixation data.

To use this model in our setup, we need to reverse this conversion process and recompute the features from token-level back to word-level, allowing us to remap the features to a different tokenizer. This is done by summing the orignal features for all tokens corresponding to a word and then distributing them across the tokens mapped to the same word in the other tokenizer. More details of this conversion are explained in Appendix A.1.3 and an example of the process in Table 8.

**Second model:** Model presented in Li & Rudzicz (2021). The code and training data are also publicly available, so we trained it following their original methodology and we adapted their code to incorporate it into our implementation. The model was trained using the ZuCo 1 (Hollenstein et al., 2018) and ZuCo 2 (Hollenstein et al., 2020a) and PROVO (Luke & Christianson, 2018) datasets. For the ZuCo datasets, 800 sentences (15.7 tokens) were provided as training data and 191 sentences (3.5k tokens) were held out for evaluation. Information about the datasets used to train these models is in Table 7. The model is based on RoBERTa (Liu et al., 2019) with a regression head on each token. This head is a linear layer that outputs five features: FFD, fixProp, GPT, TRT, and nFix (Table 1). mean absolute error (MAE) for each feature is presented in Table 6. This model has 125M parameters and remains frozen during training.

In this generative model, the conversion of word-level features to token-level features during training is done by assigning the features of a word to the first token and it is assumed that the rest of the tokens of this word do not have features. We reversed this process similarly during inference by forcing the predictions for tokens that are not the first in a word to be zero. Since the maximum sequence length for RoBERTa is 512 and we are dealing with longer sequences, we implemented a sliding window approach. We split the input into sequences of 512 tokens with a 50-token overlap. After processing, we combine the results using a linear combination.

Table 6: MAE performance of the model reported in Li & Rudzicz (2021).

| nFix | FFD | GPT | TRT | fixProp | All (Dev) |
|------|------|------|------|---------|-----------|
| 3.984 | 0.713 | 2.424 | 1.556 | 10.781 | 3.892 |

Table 7: Overview of different corpora used to train the ET features generator models.

| Corpus | Lang. | Sents. | Tokens | Subjects | Reference |
|--------|-------|--------|--------|----------|-----------|
| Dundee | EN | 2367 | 58598 | 20 | Kennedy et al. (2012) |
| Provo | EN | 189 | 2659 | 84 | Luke & Christianson (2018) |
| ZuCo 1 | EN | 300 | 6588 | 12 | Hollenstein et al. (2018) |
| ZuCo 2 | EN | 349 | 6828 | 18 | Hollenstein et al. (2020b) |
| Geco | EN* | 2449 | - | 23 | Cop et al. (2017) |

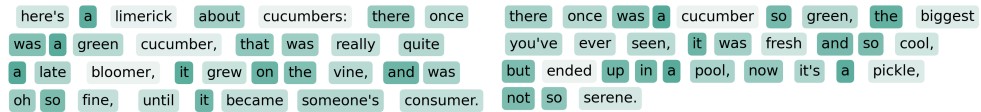

TRT per word in chosen response.     TRT per word in rejected response.

Figure 3: TRT generated by first model (Huang & Hollenstein, 2023) of the chosen and rejected response to prompt 'Create a limerick about cucumbers'. Deeper colour represents longer fixation.

### A.1.3 MAPPING ET FEATURES BETWEEN DIFFERENT TOKENIZERS

Both ET features generator models used are based on different tokenizers, which are also different from the tokenizers employed by the based models used as RM. As a result, the number of tokens $n$ in the input for the reward model and the number of tokens $w$ for the ET features may not be the same. For **GazeAdd**, to be able to combine elementwise the ET feature embedding and the text embedding, they must have the same temporal dimensions. Therefore, we need to map the ET features per token from the ET tokenizer to the tokens of the RM tokenizer. Specifically, we convert our $f_{et} \in \mathbb{R}^{w \times f}$ ($w$ is the number of tokens, and $f$ is the number of features) to $f_{et}^{mapped} \in \mathbb{R}^{n \times f}$ where $n$ is the number of tokens in the RM input. For that. we perform a mapping between the two tokenizers to obtain the *mapped features* $f_{et}^{mapped}$.

To map tokens generated by two different tokenizers, we use our *EyeTrackPy* python library that will be publicly released. First, we perform an initial mapping of tokens to the words they belong to in each tokenizer with some properties of *FastTokenizers* from the *transformers* library [8]. Then, we map words from one tokenizer to the words in the other and finally, we assume that the combination of the tokens that are mapped to a word in one tokenizer correspond to the tokens that are mapped to the word that is mapped to the initial word in the other tokenizer. Each row in Table 8, refers to a step in this process.

For each predictor, we reverse the method used to convert word-level features into token-level features (more details in Appendix A.1.2) but passing from tokens in the first one, to tokens in the second tokenizer. For example, if for the first ET features predictor models tokens $t_1, t_2$ are mapped to tokens $t_1, t_2, t_3$ in another second tokenizer, the values sum for all the tokens in the first list and distribute them equally across all the tokens in the second list: being $t_1$ (1s TRT) and $t_2$ (2s TRT) each of $t_1, t_2, t_3$ are assigned a TRT of $(1+2)/3 = 1s$. In Table 8 is represented a example of this process where row TRT(1) are the final TRT mapped for the first ET predictor and TRT(2) for the second one. Finally, because special chat tokens were removed when generating the ET features, we assign value 0 for all features in this tokens. At the time of publishing this work, some of the tokenizer functionalities needed for alignment between tokenizers were not available in Mistral 7B.

Table 8: Example of mapping TRT between two different tokenizers. TRT (1) represents the process used for the first ET predictor, and TRT(2) for the second ET predictor.

|  | **Tokenizer 1** | **Tokenizer 2** |
|---|---|---|
| Words | astrophotography | astrophotography |
| Tokens str | ['_Astro', 'photo', 'graphy'] | [Ċ, 'Ast', 'roph', 'ot', 'ography'] |
| Tokens idx | [22, 23, 24] | [23, 24, 25, 26, 271] |
| Tokens IDs | [15001, 17720, 16369] | [198, 62152, 22761, 354, 5814] |
| TRT (1) | [11.23, 11.49, 10.16] | [6.58, 6.58, 6.58, 6.58, 6.58] |
| TRT (2) | [24.53, 0, 0] | [24.53, 0, 0, 0, 0] |

---

[8]https://huggingface.co/docs/transformers/main_classes/tokenizer

### A.1.4 MODELS

As RM base models we use the pretrained checkpoint of Hugging Face for Llama 3 8B [9] (Dubey et al., 2024), Llama 3 8B-instruct [10](Dubey et al., 2024) and Mistral 7B [11] (Jiang et al., 2023a).

### A.1.5 TRAINING DETAILS

During training, the ET features predictor model remains frozen. The RM is fine-tuned on top of the open-source models using QLoRA (Dettmers et al., 2023) based on Low-Rank Adaptation (LoRA) (Hu et al., 2021), which fine-tunes select dense layers by optimizing low-rank decomposition matrices representing weight changes, instead of directly updating pre-trained weights. QLoRA introduces memory optimization techniques such as the 4-bit NormalFloat (NF4), a novel data type, to improve performance without increasing memory usage. Following Dettmers et al. (2023) we use hyperparameters: r=8, alpha=32, and dropout=0.1.

We also fine-tune the RM embedding layer, since we are adding new tokens for the chat format and special separators tokens in our **RewardConcat** method (section 4). Also, the last layer added to the RM for the scalar reward is trained from scratch without adapters. Our implementation is based in *pytorch* and we use *transformers* [12] from Hugging Face.

**Hardware.** We trained the models on servers equipped with 2x Intel Xeon Platinum 8470 CPUs, 1TB of RAM, and either 2x NVIDIA H100 (80GB) or 4x NVIDIA A100 (80GB) GPUs. We always train using only GPU at a time per each model and training times were between 20 and 50 hours depending mainly on the number of steps between model evaluations.

## B REWARD BENCHMARK

As we described in section 6, a future direction would be to train a Reward Model on a larger model with more data. It has been proven the scaling trends for the reward model; More data and a larger-size model generally improve accuracy (Touvron et al., 2023). Nevertheless, as a complement to our results, we evaluated our trained models on the dataset with the best results, OAAST1, in this RewardBench, a benchmark for Reward Models. RewardBench, proposed in Lambert et al. (2024), is a benchmark designed to evaluate the performance and safety of reward models. It consists of a set of datasets intended for measuring how reward models perform on challenging prompts across chat, reasoning, and safety domains, using a trio structure of prompt-chosen-rejected pairs. It comprises 2985 diverse tasks, each sample is formatted as a prompt with a manual or human-verified chosen and rejected completion. Due to its diversity of tasks (4 categories and 23 sub-categories) this benchmark minimizes the likelihood of overfitting. Task accuracy is calculated based on whether the chosen response receives a higher reward than the rejected response. We directly evaluate their open dataset reward-bench [13]

## C HYPER PARAMETER TUNING

We performed hyperparameter tuning for the **GazeConcat** method and the baseline, and we replicated them in the **GazeAdd** method, as testing all combinations is computationally very expensive. For each dataset, 15% is reserved for validation to perform hyperparameter tuning. The best-performing checkpoints are selected based on the lowest validation loss and are subsequently used for performance evaluation in the test split. We trained for two epochs, as described in Wang et al. (2024c) and in line with trends observed in Touvron et al. (2023) where they found that training longer can lead to over-fitting. We perform a grid search for the optimal batch size, testing {8, 16, 32} values. AdamW optimizer (Loshchilov & Hutter, 2019) is used and the learning rate is tuned within the range of {1, 5, 10, 50}e-6, inspired by the values reported in Touvron et al. (2023);

---

[9] https://huggingface.co/meta-llama/Meta-Llama-3-8B
[10] https://huggingface.co/meta-llama/Meta-Llama-3-8B-Instruct
[11] https://huggingface.co/mistralai/Mistral-7B-v0.3
[12] https://huggingface.co/docs/transformers/index
[13] https://huggingface.co/datasets/allenai/reward-bench

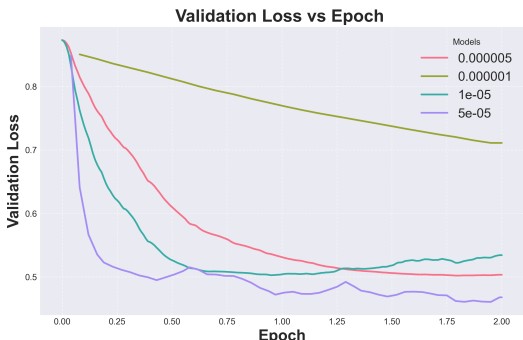

Figure 4: Validation loss with different LR on ConcatReward, batch size 8, features: $f1$ and Meta-Llama-3-8B-Instruct base model

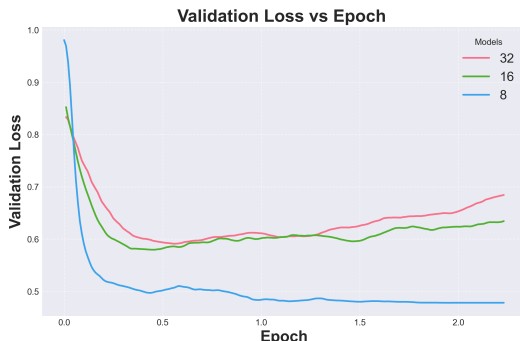

Figure 5: Validation loss with different batch size, learning rate: 5e-5, features: $f1$ and Meta-Llama-3-8B-Instruct base model

Cui et al. (2024); Wang et al. (2024c). Additionally, we explored different scheduler configuration, comparing constant, linear, and cosine with a minimum learning rate.

In general, the parameter that most affected the validation results was the learning rate. For the others, we ended up choosing values that worked well across all combinations. We achieved better results in both the baseline and the models concatenating the ET features with a learning rate of 0.0005 ( Figure 4). In Figure 6a and Figure 6b, the validation loss and learning rate with different schedulers are shown. For lower learning rates, such as 0.00001, the scheduler had little effect. However, with higher learning rates, using a scheduler helped to mitigate overfitting. In Figure 7a and Figure 7b, it represents validation loss and learning rates with a higher learning rate of 0.0005. We opted to use this 0.00005 learning rate for all experiments, employing a cosine learning rate scheduler with a minimum learning rate of 0.7, in line with other studies such as Wang et al. (2024c); Touvron et al. (2023). We did not find a significant effect of training batch size on validation accuracy, but we opted for a value of 8, which often (especially with high learning rates and without a scheduler) was the one that tended to overfit the least ( Figure 5).

**ET features projector** The PyTorch architecture of our ET features projector is shown below. $num\_features$ varies between 1, 5, and 2 (depending on the configuration used $fcomb_1$, $fcomb_{2.5}$, $fcomb_{2.2}$ subsection 5.2). $p_1, p_2$ are dropout values. Finally, after testing different combinations, we used $0.1$ and $0.3$. This model has 0.53M parameters.

Listing 1: PyTorch architecture of our gaze features projector

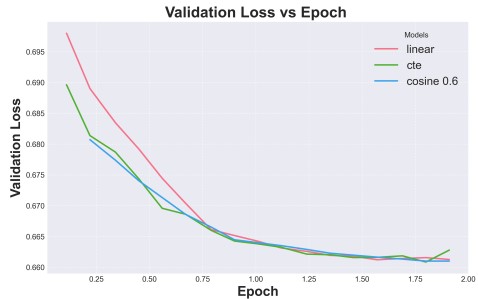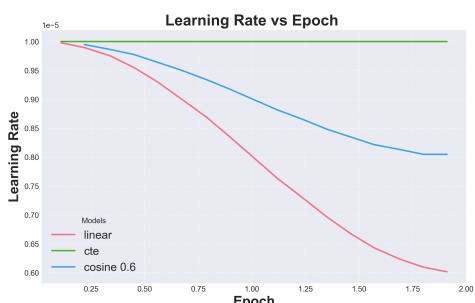

(a) Validation Loss Comparison for different LR Schedulers

(b) Learning rate comparison for different LR Schedulers

Figure 6: ConcatReward, LR: 0.00001, features: $f1$ and Meta-Llama-3-8B-Instruct base model

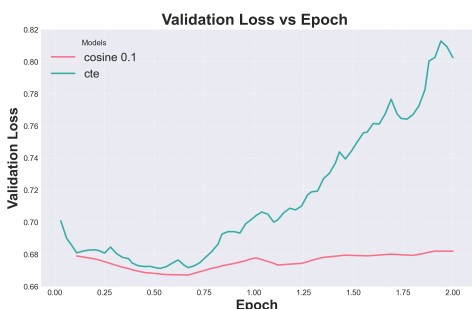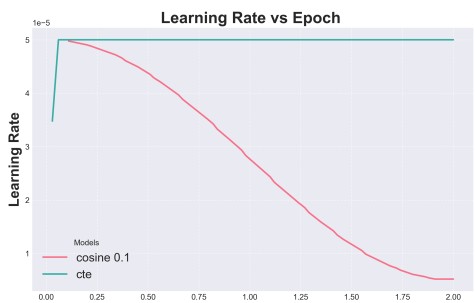

(a) Validation Loss Comparison for different LR Schedulers

(b) Learning rate comparison for different LR Schedulers

Figure 7: ConcatReward, LR: 0.00005, features: $f1$ and Meta-Llama-3-8B-Instruct base model

```
self.fixations_embedding_projector = nn.Sequential(
    nn.Linear(num_features, 128),
    nn.LayerNorm(128),
    nn.ReLU(),
    nn.Dropout(p=p_1),
    nn.Linear(128, hidden_size),
    nn.Dropout(p=p_2),
)
```

