# OpenReview forum: "Seeing Eye to AI: Human Alignment via Gaze-Based Response Rewards for Large Language Models"
_ICLR.cc/2025/Conference — ICLR 2025 Poster_

### Official Review · Reviewer_WBeE · 2024-11-04

**Soundness:** 2
**Presentation:** 2
**Contribution:** 3
**Rating:** 6
**Confidence:** 3

**Summary:**

This work incorporates features extracted from eye-tracking prediction models into reward models for RLHF. They find that adding eye-tracking information improves reward modeling accuracy on two reward modeling datasets. In addition, the authors also perform ablation studies to study what type of eye-tracking features lead to the highest reward modeling accuracy.

**Strengths:**

The idea behind incorporating eye-tracking features to reward models is quite original, and to my knowledge, has not been previously explored. The authors are also able to demonstrate accuracy gains in reward modeling after adding eye-tracking features, which brings a positive impact to RLHF that is widely used in current LLMs.

**Weaknesses:**

I find two main weaknesses that prevent me from recommending this paper for acceptance, but that could be feasibly addressed by the authors. First, it is difficult for me to assess whether the improvements on reward modeling reported in the paper are significant. According to Table 2, the test sets used for reward model evaluation are quite small (416 and 364 examples) and I did not find information on the statistical significance of the differences in results. I am also skeptical whether the improvements in scores would be robust to different random seeds and other parameters. It would be beneficial to add statistical significance tests and further discussion on how you ensured your findings are robust.
The second weakness is the lack of motivation in using eye-tracking data. There are a few points the authors allude to in the introduction (see some of my questions below), but it would be very helpful to explain the intuition on why eye-tracking data would have a connection or be correlated with human preferences. Intuitively, I would think that people would spend more time gazing at sentences that are difficult to parse for example, rather than having a preference.

**Questions:**

- L.46-48 Could you elaborate how eye-tracking data mitigates the problems you cited in human feedback for RLHF? Would the eye-tracking features not also reflect biases from humans from which the ET models were trained on, or their domain expertise?
- L.48-49 How is scalable oversight relevant to the problem studied in this paper?
- L. 73 It's unclear what you mean by "better temporal and spatial resolution", could you please clarify?
- L. 74 What is the connection to attention patterns and human preference for reward modeling?
- S. 3.1 It would be helpful to describe more in this section what these ET models are trained on, the accuracy of these ET models.
- I would be curious to see results, or if not feasible a discussion, on how your GazeReward framework compares to if we used eye-tracking data directly collected on the preference prompts instead of features generated by an ET prediction model
- Please also discuss the ethical impact of this work, mainly to address concerns that incorporating an ET model may amplify biases present in the ET models
- Suggestion: make the text in your figures bigger

---

> ### Author Response · Authors · 2024-11-18
> **Response to Official Review of Submission11182 by Reviewer WBeE**
>
> * **W1** First, it is difficult for me to assess whether the improvements on reward modeling reported in the paper are significant. According to Table 2, the test sets used for reward model evaluation are quite small (416 and 364 examples) and I did not find information on the statistical significance of the differences in results. I am also skeptical whether the improvements in scores would be robust to different random seeds and other parameters. It would be beneficial to add statistical significance tests and further discussion on how you ensured your findings are robust.
>
> Thank you for this comment. Our experiments involve the rigorous testing of six baselines against six models each (with the exception of Mistral which has three models). In total, we trained and evaluated 6 baselines and 30 models. For 28 out of those 30 models, we have demonstrated significant performance gains compared to their respective baseline. Regarding the robustness of results across different parameters or seeds, and taking into account the computational requirements, we wish to highlight that our methodology implements standard practices in the field, as demonstrated in prior works [Cui et al., 2024; Wang et al., 2024c; Touvron et al., 2023; Köpf et al., 2023]. These studies also report results based on a single seed for models of comparable scale. Currently, our computational resources do not allow us to retrain all models. However, we are in the process of training one configuration with a different seed and anticipate sharing those results before the rebuttal period concludes. Conducting a comprehensive evaluation across multiple seeds for all models would require approximately one month, and we would be prepared to include this analysis in the camera-ready version of the paper, should it be accepted.
>
> We have calculated the mean and standard deviation of the differences in validation accuracy between each model variant and the baseline, using the 5 checkpoints closest (in time) to the highest accuracy. We conducted two-sample t-tests calculating test statistics (\(t\)) and p-values (\(p\)) for each comparison. The next two tables show all results to be statistically significant. To control for Type I errors (i.e. false positive cases) due to the multiple comparisons, we applied the two-stage Benjamini-Hochberg (fdr\_tsbh) method to adjust p-values and control the false discovery rate (FDR).
>
> **Table: Mean and standard deviation of the difference in accuracy for each combination and the baseline.**
> The mean is based on the 5 checkpoints in validation for the oasst1 dataset. * indicates significance (**p-values < 0.01 and *p-values < 0.05).
>
> |          |                   | **LLaMA-Instruct**   | **LLaMA**           | **Mistral**         |
> |----------|-------------------|----------------------|---------------------|---------------------|
> | **Concat** | $fcomb_{1}$       | 8.5 ± 0.8**       | 4.5 ± 1.0*        |   2.3 ± 0.2*       |
> |          | $fcomb_{2.5}$     | 4.7 ± 1.1*        | 7.6 ± 0.7*       |   -0.6 ± 1.5*       |
> |          | $fcomb_{2.2}$     | 5.7 ± 1.9*        | 4.7 ± 3.1*        |   -0.3 ± 1.8*       |
> | **Add**   | $fcomb_{1}$       | 1.0 ± 0.3*        | 2.6 ± 0.3*        |       -         |
> |          | $fcomb_{2.5}$     | 2.4 ± 0.5*        | 6.3 ± 0.2*        |       -         |
> |          | $fcomb_{2.2}$     | 12.6 ± 0.2**      | 6.0 ± 0.3*        |       -         |
>
>
> **Table: Mean and standard deviation of the difference in accuracy for each combination and the baseline.**
> The mean is based on the 5 checkpoints in validation for the helpsteer2 dataset. * indicates significance (**p-values < 0.01 and *p-values < 0.05).
>
> |                |                  | **LLaMA-Instruct**   | **LLaMA**           | **Mistral**         |
> |----------------|------------------|----------------------|---------------------|---------------------|
> | **Concat**     | $fcomb_{1}$      | 8.5 ± 0.8 **     | 4.5 ± 1.0 *       | 2.3 ± 0.2 *       |
> |                | $fcomb_{2.5}$    | 4.7 ± 1.1 *        | 7.6 ± 0.7 *       | -0.6 ± 1.5 *      |
> |                | $fcomb_{2.2}$    | 5.7 ± 1.9 *        | 4.7 ± 3.1 *       | -0.3 ± 1.8 *      |
> | **Add**        | $fcomb_{1}$      | 1.0 ± 0.3 *        | 2.6 ± 0.3 *       | -                  |
> |                | $fcomb_{2.5}$    | 2.4 ± 0.5 *        | 6.3 ± 0.2 *       | -                  |
> |                | $fcomb_{2.2}$    | 12.6 ± 0.2 **    | 6.0 ± 0.3 *       | -                  |

---

> ### Author Response · Authors · 2024-11-18
>
> **References:**
>
> Ganqu Cui, Lifan Yuan, Ning Ding, Guanming Yao, Bingxiang He, Wei Zhu, Yuan Ni, Guotong Xie, Ruobing Xie, Yankai Lin, Zhiyuan Liu, and Maosong Sun. UltraFeedback: Boosting Language Models with Scaled AI Feedback, July 2024. URL http://arxiv.org/abs/2310.01377 [cs].
>
>
> Zhilin Wang, Yi Dong, Olivier Delalleau, Jiaqi Zeng, Gerald Shen, Daniel Egert, Jimmy J. Zhang, Makesh Narsimhan Sreedhar, and Oleksii Kuchaiev. HelpSteer2: Open-source dataset for training top-performing reward models, June 2024c. URL http://arxiv.org/abs/2406.08673, arXiv:2406.08673 [cs].
>
>
> Hugo Touvron, Louis Martin, et al. Llama 2: Open Foundation and Fine-Tuned Chat Models, July 2023. URL http://arxiv.org/abs/2307.09288, arXiv:2307.09288 [cs].
>
>
> Andreas Kopf, Yannic Kilcher, Dimitri von R\"{u}tte, Sotiris Anagnostidis, Zhi-Rui Tam, Keith Stevens, Abdullah Barhoum, Nguyen Minh Duc, Oliver Stanley, Rich\'{a}rd Nagyfi, Shahul ES, Sameer Suri, David Glushkov, Arnav Dantuluri, Andrew Maguire, Christoph Schuhmann, Huu Nguyen, and Alexander Mattick. OpenAssistant Conversations – Democratizing Large Language Model Alignment, April 2023. URL http://arxiv.org/abs/2304.07327. arXiv:2304.07327 [cs].

---

> ### Author Response · Authors · 2024-11-18
>
> * **W2** The second weakness is the lack of motivation in using ET data. There are a few points the authors allude to in the introduction (see some of my questions below), but it would be very helpful to explain the intuition on why ET data would have a connection or be correlated with human preferences. Intuitively, I would think that people would spend more time gazing at sentences that are difficult to parse for example, rather than having a preference. **Q4** L. 74 What is the connection to attention patterns and human preference for reward modeling?
>
> A substantial body of research has explored oculomotor behaviour in the context of reading comprehension and broader reading tasks, offering valuable insights into how attention patterns relate to human preferences and reward modeling. For instance, Adams et al. (2015) demonstrate that oculomotor behavior reflects precision-weighted prediction error minimization within the framework of active inference. This connects to preference modeling as follows: attention patterns revealed through eye movements provide a direct window into what information humans prioritise when making decisions - essentially encoding implicit preferences through selective sampling of sensory data. Just as Adams' model shows how precision expectations guide pursuit of relevant visual targets, these same mechanisms likely shape how humans allocate attention when evaluating options to arrive at preference judgments. Similarly, models like UniAR (Li et al., 2023) further highlight how visual attention patterns can predict explicit human feedback, such as aesthetic judgments or subjective preferences.
>
> The key insight here is that precision estimates can be reliably inferred from observed eye movements. This suggests that attention patterns may serve as objective behavioural markers for validating and refining reward models. By analysing how attention allocation during evaluation correlates with expressed preferences, we can better understand and model the computational mechanisms underlying human reward learning and decision-making. We can include a section in the Appendix if it would clarify the motivation.
>
> **References:**
>
> Adams, R. A., Aponte, E., Marshall, L., & Friston, K. J. (2015). Active inference and oculomotor pursuit: the dynamic causal modelling of eye movements. Journal of neuroscience methods, 242, 1–14. https://doi.org/10.1016/j.jneumeth.2015.01.003
>
>
> Li, P., He, J., Li, G., Bhargava, R., Shen, S., Valliappan, N., Liang, Y., Gu, H., Ramachandran, V., Farhadi, G., Li, Y., Kohlhoff, K., Navalpakkam, V. (2023). UniAR: A Unified model for predicting human Attention and Responses on visual content.

---

> > ### Author Response · Authors · 2024-11-18
> >
> > * **Q1** L.46-48 Could you elaborate how ET data mitigates the problems you cited in human feedback for RLHF? Would the ET features not also reflect biases from humans from which the ET models were trained on, or their domain expertise?
> >
> > Thank you for this comment. In our paper, we make two separate references to the pervasive challenges in regard to human feedback. The first one concerns data annotation issues ( "*Obtaining high-quality feedback from human annotators, usually provided after examining a model response, suffers from several caveats (Casper et al., 2023). For instance, low inter-annotator agreement can result in inconsistent evaluations of the same model output due to varying interpretations, domain expertise, or biases."* ). Here, we do not claim that ET data mitigates any of these problems -- instead, we cite them to explain in the follow up paragraph why the research community has been increasingly turning to LLMs as a form of AI-driven feedback (RLAIF), to address some of the issues with RLHF. Therefore, the issues addressed by RLAIF are data-specific issues that are inherent to human feedback methods. However, RLAIF methods, although scalable and cost-effective, introduce a new challenge: that of human alignment. Specifically, this challenge concerns how the user data are operationalised into human alignment techniques and are affected by the quality of the underlying reward model. With respect to this latter challenge, we argue that generative models (trained on large volumes of individual user data) can provide synthetic ET data and act as a ``surrogate'' user that exhibits as "average" behaviour. This average behaviour encapsulates the behaviour of the users the model was trained on and, in that respect, can correct the biases of user-annotated data.
> >
> > We would like to clarify that our paper comments on human feedback challenges and, specifically, annotation issues (*Obtaining high-quality feedback from human annotators ... can result in inconsistent evaluations of the same model output due to varying interpretations, domain expertise, or biases.}*) to give a broader context about why researchers have turned to AI feedback, among other approaches, and managed to address mainly the scalability issue. Here, we propose that ET data can have a complementary role in also improving the alignment of the feedback. Specifically, we position ET features synthesized by generative models that are trained on large datasets as a cost-efficcient solution to the broader human alignment challenges. The main advantage of ET-based approaches lies in their ability to act as a surrogate for an ``average'' user, encapsulating aggregate user behavior, while minimizing the individual variations that typically skew user-annotated data.
> >
> > While ET models inevitably reflect aspects of their training data, their strength lies in leveraging diverse inputs to generalize and counteract idiosyncratic human biases rather than amplify them. This perspective is central to our argument that ET-enhanced approaches can complement existing methods and refine alignment techniques. Recent research supports this view. For instance, Li et al. (2023) demonstrated how a single ML model can predict rich human feedback on generated images, including text-image misalignment, aesthetic quality, and problematic regions with artifacts, along with explanations.
> >
> > Although a comprehensive review of supporting literature extends beyond our paper's scope, numerous studies support how ET-enhanced methods may effectively complement existing alignment techniques, while addressing the concerns about bias and expertise that you have raised. Of particular relevance is the field of computational phenotyping, which provides additional validation for our approach. For example, deep temporal models can simulate synthetic subjects performing experimental paradigms. These generative models can be fine-tuned to match empirical subject performance, enabling simulations of human eye movements during tasks like reading or decision-making under uncertainty. This process illustrates how such models accumulate beliefs or information at multiple levels -- from individual letters to words and sentences over varying time periods.

---

> > > ### Author Response · Authors · 2024-11-18
> > >
> > > **References:**
> > > Computational Phenotyping in Psychiatry: A Worked Example
> > > Philipp Schwartenbeck, Karl Friston
> > > eNeuro 18 July 2016, 3 (4) ENEURO.0049-16.2016; DOI: 10.1523/ENEURO.0049-16.2016
> > >
> > > Comparing visual discrimination and detection: the special status of ‘no’ responses Journal of Vision ( IF 2 Submission Guide >) Pub Date: 2019-09-06 , DOI:10.1167/19.10.142c Matan Mazor,Lucie Charles,Karl J. Friston,Stephen M. Fleming
> > >
> > > Allen, M., Levy, A., Parr, T., \& Friston, K. J. (2022). In the Body's Eye: The computational anatomy of interoceptive inference. PLoS computational biology, 18(9), e1010490. https://doi.org/10.1371/journal.pcbi.1010490
> > >
> > > Li, P., He, J., Li, G., Bhargava, R., Shen, S., Valliappan, N., Liang, Y., Gu, H., Ramachandran, V., Farhadi, G., Li, Y., Kohlhoff, K., & Navalpakkam, V. (2023). UniAR: A Unified model for predicting human Attention and Responses on visual content.
> > >
> > > Vincent, P., Parr, T., Benrimoh, D., \& Friston, K. J. (2019). With an eye on uncertainty: Modelling pupillary responses to environmental volatility. PLoS computational biology, 15(7), e1007126. https://doi.org/10.1371/journal.pcbi.1007126
> > >
> > > Perrinet, L. U., Adams, R. A., \& Friston, K. J. (2014). Active inference, eye movements and oculomotor delays. Biological cybernetics, 108(6), 777–801. https://doi.org/10.1007/s00422-014-0620-8
> > >
> > > Brodersen, K. H., Penny, W. D., Harrison, L. M., Daunizeau, J., Ruff, C. C., Duzel, E., Friston, K. J., \& Stephan, K. E. (2008). Integrated Bayesian models of learning and decision making for saccadic eye movements. Neural networks : the official journal of the International Neural Network Society, 21(9), 1247–1260. https://doi.org/10.1016/j.neunet.2008.08.007
> > >
> > > Adams, R. A., Aponte, E., Marshall, L., \& Friston, K. J. (2015). Active inference and oculomotor pursuit: the dynamic causal modelling of eye movements. Journal of neuroscience methods, 242, 1–14. https://doi.org/10.1016/j.jneumeth.2015.01.003

---

> > > > ### Author Response · Authors · 2024-11-18
> > > >
> > > > * **Q2**. L.48-49 How is scalable oversight relevant to the problem studied in this paper?
> > > >
> > > > The concept of ``scalable oversight'' is primarily referenced in our paper to offer a broader context of challenges in data generation, rather than being central to the research problems we address. We mention it to frame the shift toward more efficient AI-driven feedback mechanisms (RLAIF). While traditional oversight approaches often require substantial human involvement and resources, RLAIF offers a scalable and cost-efficient alternative by leveraging LLM agents as imperfect but scalable proxies for generating feedback. However, it is important to note that our primary focus is not on solving the scalable oversight problem itself, but rather on developing practical solutions for efficient feedback generation within resource constraints.
> > > >
> > > > * **Q3** L. 73 It's unclear what you mean by "better temporal and spatial resolution", could you please clarify?
> > > >
> > > > Thank you for this question. By ``better temporal and spatial resolution'', we were referring to the advantages of using objective physiological signals, such as \acrshort{et}, that provide more fine-grained information compared to other feedback methods like questionnaire data, manual labels, or behavioural data. ET devices -- for instance the Tobii Pro Fusion 250Hz -- can capture data at a sampling frequency of up to 250Hz, with a mean latency under 13ms. This provides a much higher temporal resolution compared to relying on user feedback or annotations, which typically only provide one data point per prompt. Regarding spatial resolution, ET data comes with a precision of 0.04° RMS and accuracy of 0.3°, and eye image data stream at approximately 4 Hz (one image per eye). This allows for more granular spatial tracking of user attention and focus on the screen, compared to coarser behavioural or subjective feedback. We have expanded on these points in the paper to better explain the benefits of the physiological ET signals over other data sources.
> > > >
> > > > * **Q5** S. 3.1 It would be helpful to describe more in this section what these ET models are trained on, the accuracy of these ET models.
> > > >
> > > > We provide more details about the training of these models in section 4.1 - ET prediction models. Due to space limitations, part of this information is also in Appendix A1.2, where we have more information about the datasets used to train each model. We have also added (now in red for review) the performance metrics for these models in the Appendix, due to space issues.
> > > >
> > > > * **Q6** I would be curious to see results, or if not feasible a discussion, on how your GazeReward framework compares to if we used ET data directly collected on the preference prompts instead of features generated by an ET prediction model.
> > > >
> > > > In section 6.1, under the Data subsection (L510-513), we discuss this: *``Future work could benefit from directly collecting ET data specifically for LLM-generated responses, to offer insights into human reading comprehension and information processing of prompts, which could further improve model performance.''* We believe that the availability of specifically annotated ET data for this task, or data generated by models trained on an ET corpus tailored for this purpose, could yield even greater improvements in the accuracy of reward models than proposed here. To the best of our knowledge, no such ET datasets currently exist in the literature for this application. In fact, this work represents the first exploration of ET data to enrich preference models for human alignment. Specifically, our study demonstrates the feasibility and justifies the need for future efforts to create dedicated ET datasets. Such datasets would allow a more comprehensive investigation into the potential of integrating ET data into models designed to measure user preferences, thereby advancing human alignment research.
> > > >
> > > > * **Q7** Please also discuss the ethical impact of this work, mainly to address concerns that incorporating an ET model may amplify biases present in the ET models.
> > > >
> > > > Modeling any aspect of human behavior should adhere to ethical guidelines on data collection and applications, and be conducted transparently, including clarifying the limitations of the model when replicating human preferences. Taking these considerations into account, we have added it in the Impact Statement.
> > > >
> > > > * **Q8** Suggestion: make the text in your figures bigger
> > > >
> > > > Thank you for the suggestion; we will enlarge all text elements in our figures to ensure better readability.

---

> > > > > ### Author Response · Authors · 2024-11-26
> > > > >
> > > > > We would like to thank the reviewer for the time you have invested in reviewing our paper and for the initial comments. Tomorrow is the last day we will be able to edit the PDF, so we would appreciate any feedback on our current responses and modifications, as well as any clarifications you might require.

---

> > > > > > ### Comment · Reviewer_WBeE · 2024-12-03
> > > > > >
> > > > > > Thank you for your thorough response, I appreciate it! The additional report of statistical significance of your results is especially helpful, and I have updated my rating.

---

### Official Review · Reviewer_TgMi · 2024-11-05

**Soundness:** 3
**Presentation:** 3
**Contribution:** 3
**Rating:** 6
**Confidence:** 4

**Summary:**

The paper introduces GazeReward, a framework that incorporates eye-tracking data into the reward modeling phase of RLHF to better align LLMs with human preferences. This represents a novel solution to the reward modeling task with extra information from eye-tracking features. Experimental results also show the effectiveness of the proposed approach with an impormvent in reward modeling accuracy.

**Strengths:**

+ Eye-tracking is being introduced as a novel feature to assist reward modeling.
+ Substantial performance improvements with the GazeReward framework

**Weaknesses:**

+ The experiments show that eye-tracking signals can improve the accuracy of reward modeling. As the reward model is used for RLHF processes like PPO and Best-of-n sampling, I would suggest the authors add relevant experiments to show the improvement in this down-stream application of the reward model.
+ What is the size of eye-tracking feature extractor? If the size of the eye-tracking component is not negligible, it might not be a fair comparison between the original model the model with eye-tracking feature extractor.

**Questions:**

See weaknesses above.

---

> ### Author Response · Authors · 2024-11-18
> **Response to Official Review of Submission11182 by Reviewer TgMi**
>
> * **W1**  The experiments show that ET signals can improve the accuracy of reward modeling. As the reward model is used for RLHF processes like PPO and Best-of-n sampling, I would suggest the authors add relevant experiments to show the improvement in this down-stream application of the reward model.
>
> We agree that it would be valuable to demonstrate the impact of the ET-enhanced reward model on downstream RLHF processes like PPO and Best-of-n sampling. Unfortunately, due to computational constraints, we were not able to include those additional experiments in this work. We hope the presented experiments effectively demonstrate the model's potential within our current resource limits and reserve the investigation of performance gains in these downstream applications in follow-up research.
>
>
> * **W2**  What is the size of ET feature extractor? If the size of the ET component is not negligible, it might not be a fair comparison between the original model the model with ET feature extractor.
>
> Thank you for this feedback. To address your concerns about the model size comparison, we included the following information in the Appendix A.1.2: The first ET feature extractor has 17.5M parameters, while the second one has 125M parameters. More importantly, the ET component remains frozen during fine-tuning, so its parameters are not updated for the reward modeling task. We believe this makes the comparison with the baseline fair, as the overall model capacity is not significantly different. While the reward models we use are quite large (7-8 billion parameters), we agree that the relative size of the ET component is still worth considering.
>
>
> We have expanded our  "Discussion" section to acknowledge this as a potential factor to keep in mind when interpreting the results.

---

> > ### Author Response · Authors · 2024-11-26
> >
> > We would like to thank the reviewer for the time you have invested in reviewing our paper and for the initial comments. Tomorrow is the last day we will be able to edit the PDF, so we would appreciate any feedback on our current responses and modifications, as well as any clarifications you might require.

---

### Official Review · Reviewer_sHLc · 2024-11-06

**Soundness:** 3
**Presentation:** 4
**Contribution:** 3
**Rating:** 8
**Confidence:** 3

**Summary:**

The paper examines the possibility of using eye-tracking data to train language reward models. The authors integrate synthetic eye tracking data from existing models into the reward model using two strategies: concatenation or addition of eye tracking features. They test their methods across multiple eye tracking models and LLMs and show that eye tracking data can improve reward models at a small to medium scale.

**Strengths:**

S1. The idea of using eye tracking data as reward model signal is exciting, interesting, and well-explored in this work. The paper analyzes several methods of integration and discusses the pros and cons of (synthetic or real) eye tracking data.

S2. The paper is beautifully written and clearly lays out prior work in both reward modeling and eye tracking for NLP. It was very easy to follow.

S3. The appendix reports details on hyperparameter tuning, implementations, and modeling assumptions. I believe it would be easy to replicate this work with the details provided.

**Weaknesses:**

W1. Very minor, but I think section 2+5 could be better structured, perhaps by moving them both into section 2. I was a bit surprised on my first read to see a related work section near the end, since section 2 discusses much of the reward modeling literature; I think it makes sense to put these in the same spot, and I liked the placement near the beginning.

W2. Because the datasets are somewhat constrained, it's not clear that this would provide consistent gains over existing reward models in a large-scale setting (could this be a useful signal only in small data regimes?). I don't think the paper necessarily needs to provide evidence against this concern, as this might require excessive amounts of compute, but I think more discussion of this possibility would help temper the claims in the work.

**Questions:**

Q1. You don't scale this approach to larger models/datasets (understandably!). Do you foresee any potential challenges of scaling this approach other than compute cost? Do you believe this data would be best used to fully replace existing methods for reward modeling, or in conjunction?

Q2. Somewhat related to the above-- do you have a sense of any error category differences between RMs trained with eye tracking data and without?

---

> ### Author Response · Authors · 2024-11-18
> **Response to Official Review of Submission11182 by Reviewer sHLc**
>
> * **W1** Very minor, but I think section 2+5 could be better structured, perhaps by moving them both into section 2. I was a bit surprised on my first read to see a related work section near the end, since section 2 discusses much of the reward modeling literature; I think it makes sense to put these in the same spot, and I liked the placement near the beginning.
>
>
>
> Thank you for the helpful suggestion. We agree that moving section 5 after section 2 will improve the flow and clarify the structure. However, we believe keeping  the "Related Work" separate from "Background" serves a purpose. In the revised version, we have applied this change, per your recommendation.
>
>
>
> * **W2**  Because the datasets are somewhat constrained, it's not clear that this would provide consistent gains over existing reward models in a large-scale setting (could this be a useful signal only in small data regimes?). I don't think the paper necessarily needs to provide evidence against this concern, as this might require excessive amounts of compute, but I think more discussion of this possibility would help temper the claims in the work.
>
>
> You raise a valid point that the dataset constraints may limit the generalizability of the findings. While the paper does not provide definitive evidence against this concern, we have expanded the "Discussion" section to acknowledge this as a potential challenge and suggest directions for future research to explore the approach's performance in larger-scale settings. We hope this helps provide a more balanced perspective.
>
>
> * **Q1** You don't scale this approach to larger models/datasets (understandably!). Do you foresee any potential challenges of scaling this approach other than compute cost? Do you believe this data would be best used to fully replace existing methods for reward modeling, or in conjunction?
>
>
> Thank you for the insightful feedback. We do not foresee any major challenges in scaling our approach, beyond the computational cost. The use of gemerative models for producing ET features should allow the solution to scale to larger datasets and models. Moreover, we believe the ET data can enhance existing reward modeling methods, rather than fully replacing them. The goal is to incorporate this additional signal to better capture user preferences, while still leveraging the strengths of current techniques. We have expanded the "discussion" in the paper to clarify this perspective.
>
>
> * **Q2**  Somewhat related to the above-- do you have a sense of any error category differences between RMs trained with ET data and without?
>
>
>
> The datasets we examined include both preferred and non-preferred annotations, typical of those used in human alignment tasks. Although we did not directly compare error categories between reward models trained with and without ET data, we believe that using data more closely aligned with the ET data used in the predictor model's training would enhance overall accuracy. In other words, we believe that preference data more similar to the data used for training the ET predictors could benefit from greater gains in performance. However, you raise a valid point and we believe that, as the fixation generation models and available datasets continue to evolve, the potential issues you highlighted can be further addressed.

---

> > ### Comment · Reviewer_sHLc · 2024-11-18
> >
> > Thanks for the reply! I'm also glad to see the additional statistical testing results in the response to reviewer `WBeE`. I won't update my score as it's already quite positive, but I think this is a good paper!

---

### Author Response · Authors · 2024-12-02
**Final comment by authors**

As a final comment on our paper review, we’d like to summarize the clarifications provided during the rebuttal:
* Reviewer sHLc: Expanded the discussion on large-scale settings, models, and datasets.
* Reviewer TgMi: Added details on ET feature extractor models and explored the effects of model size.
* Reviewer WBeE: Included tables showing statistical significance, presented state-of-the-art examples with similar training methods, increased training seeds across all 36 runs, and committed to reporting these results upon acceptance. We also addressed the motivation of ET signals by citing relevant literature and updated the Impact Statement accordingly.

We have requested feedback from both TgMi and WBeE on our responses. As the rebuttal period concludes, we welcome any final clarifications and kindly ask that reviewers consider our updates when finalizing their scores. Thank you for your time and consideration.

---

### Meta-Review · Area_Chair_b95H · 2024-12-19

**Metareview:**

This paper introduces a novel framework that integrates eye-tracking data as implicit feedback into the reward model for aligning LLMs with human expectations, called GazeReward. The reviewers appreciate the novelty of the approach and the performance gains it can achieve by incorporating gaze data. However, they also expressed concerns about the limited scale of the experiments, the lack of clear justification for using eye-tracking data, the absence of downstream evaluation, and the lack of statistical analysis.

**Additional Comments On Reviewer Discussion:**

The authors respond to the reviewers' concerns and questions but most reviewers did not engage in discussion. While questions were clarified and some concerns such as statistical significance are addressed, others such has the concern about limited evaluation are not adequately addressed. Overall, however, the reviewers believe this work is interesting.

---

### Decision · Program_Chairs · 2025-01-22

Accept (Poster)